# Revisiting the factor structure of the EURO-D, a depressive symptomatology scale: An exploratory graph analysis in 26 European countries

exploratory graph analysis; validity; factor structure; EURO-D; depression

**Corresponding author:**
Zaira Torres;
Email: zaira.torres@uv.es

Mireia Abella, José M. Tomás, Zaira Torres , Aitana Sanz and Irene Fernández

Department of Methodology of the Behavioral Sciences, University of Valencia, Spain

## Abstract

Despite the widely use and multiple validations of the EURO-D scale, its factor structure is still under debate. Exploratory Graph Analysis (EGA), a novel network psychometric method, offers a promising approach to examining dimensionality. Methodology: 45,390 participants (mean age = 71.27, 57.4% women) from 26 European countries. The sample was randomly split into a derivation sample ($n$ = 22,823) and a cross-validation sample ($n$ = 22,567). EGA was applied to the derivation sample to determine the structure of the EURO-D scale, utilizing two estimation methods: Graphical Least Absolute Shrinkage and Selection Operator (GLASSO) and Triangulated Maximally Filtered Graph (TMFG). The identified factor structures were then tested via Confirmatory Factor Analysis (CFA) in the cross-validation sample for model fit. Results: EGA consistently revealed a two-factor structure with minor differences in the placement of suicidality and fatigue items across estimation methods. CFA results confirmed an adequate model fit for both solutions. Conclusion: This study combines exploratory (EGA) and confirmatory (CFA) approaches, supporting a two-factor structure for the EU-RO-D scale with alternative placements for fatigue and suicidality items. Results are discussed in contrast to previous studies reporting two and three-factor solutions with different assignments of these items.

## Impact statement

Understanding depression in older adults is crucial for public health, especially as Europe's population continues to age. The EURO-D scale is one of the most widely used tools to assess depressive symptoms in older adults, and it shapes a large body of research; however, past studies have produced inconsistent findings about the structure of the scale, raising questions about how well it captures depression symptomatology.

This study provides new evidence about the underlying structure of the EURO-D scale by applying Exploratory Graph Analysis (EGA), a modern network-based method that offers a data-driven way to detect dimensions in psychological questionnaires. Using data from more than 46,000 adults aged 50 years or more across 26 European countries, this research shows that EURO-D consistently reflects two main dimensions of depressive symptoms: affective suffering (emotional distress) and lack of motivation (reduced interest, energy and engagement). The findings also clarify that the items of "suicidality" and "fatigue" show instability, information that is highly valuable for researchers and clinicians interpreting EURO-D scores.

Beyond methodological advances, this work strengthens the foundation for cross-national studies of aging and mental health. More accurate knowledge of what the EURO-D measures supports better prevalence estimates and enhances the validity of the scale. By reinforcing the validity of the EURO-D while identifying areas for refinement, these findings can inform future mental health assessments, survey design, and policy efforts aimed at addressing depression in aging societies.



## Introduction

Morbidity attributable to mental disorders worldwide has been growing steadily in recent years, while the quality of mental health services is worse than the quality of physical health services (Patel et al., 2018). According to GBD's (2021) Mental Disorders Collaborators, depression affects over 350 million people, including more than 49 million in Europe. Depression impacts mental and physical health, and can potentially lead to disability, suicide and death (GBD, 2021).

The estimates of depression show significant geographical variation (Abdoli et al., 2022; Núñez-Cortés et al., 2025). Several studies based on European data suggest that depression prevalence across Europe is somewhere between 5% and 10%, with important differences among countries (Lim et al., 2018; Reibling et al., 2017). Using data from the European Health Interview

Survey (EHIS-2) across 27 European countries, De la Torre et al. (2021) found that the overall prevalence of depressive disorder was 6.38%, with prevalence grossly varying across countries, from 2.58% to 10.33%. Moreover, in 2022, regional differences in depression prevalence were observed across Europe: Western Europe reported a rate of 29.8%, Eastern Europe 28.4%, Southern Europe 27.0%, and Northern Europe 24.1% (Rodríguez-Donate et al., 2024).

Depressive symptomatology increases with old age (Yang et al., 2023), and the proportion of adults worldwide suffering from depression is expected to become the world's largest disease burden by 2030 (WHO, 2023). The study of Abdoli et al. (2022) that analyzed 20 studies showed that the prevalence of depression at old age was estimated to be 13.3% in general. Additionally, Zenebe et al. (2021) estimated the prevalence of depression at old age at 31.74% in a meta-analysis. By gender, older women report a higher prevalence of depression and depressive symptoms than older men (Abdoli et al., 2022; Cheung and Mui, 2023; Rodríguez-Donate et al., 2024).

Thus, depressive symptomatology and depression in older people are not trivial problems, and therefore, a good measurement of its occurrence is of vital importance for adequate treatment of the problem. There are several well-established scales to measure depressive symptomatology that have been extensively used to measure older adults' symptoms of depression: the SHORT-CARE scale (Gurland et al., 1984); the Geriatric Depression Scale (GDS), and its shorter versions as GDS–S (Sheikh and Yesavage, 1986), GDS-4 (van Marwijk et al., 1995) and GDS-5 (Hoyl et al., 1999); and the EURO-D scale (Prince et al., 1999). Among these scales, the EURO-D is of particular interest to us, as it is used in longitudinal panel surveys of older adults, such as the Survey of Health, Ageing, and Retirement in Europe (SHARE).

The EURO-D scale was developed to harmonize several existing depression measures (Prince et al., 1999), including the Geriatric Mental State-Automated Geriatric Examination for Computer-Assisted Taxonomy (GMS-AGECAT) (Copeland et al., 1986), SHORT-CARE (Gurland et al., 1984), the Center for Epidemiologic Studies Depression Scale (CES-D) (Radloff, 1977), the Zung Self-Rating Depression Scale (ZSDS) (Zung, 1965), and the Comprehensive Psychopathological Rating Scale (CPRS) (Asberg et al., 1978). In the original study with 21,724 participants aged 65 years and over from 14 European countries, Prince et al. (1999) reported evidence of reliability, although evidence of dimensionality of the scale was not so clear. Results yielded a two-factor solution of depressed affect and lack of motivation in some countries, while a third factor of either somatization, irritability or guilt emerged in other countries.

Castro-Costa et al. (2007) replicated a similar two-factor structure, with some modifications, using data from 10 European countries through Confirmatory Factor Analysis (CFA). Further work by Guerra et al. (2015) in Cuba, the Dominican Republic, Puerto Rico, Peru, Venezuela, Mexico, China, India, and Nigeria, using Principal Component Analysis (PCA), identified either two or three factors. Across sites, the first two factors explained 36.4–45.8% of the cumulative variance, and a third factor accounted for an additional 8.4–9.3%. Depression and tearfulness typically define the first factor, while interest and enjoyment define the second. In Venezuela, the third factor was dominated by enjoyment and interest; in Nigeria, the third factor was dominated by depression and crying. Elsewhere, the third factor was loaded by a variety of items: guilt, with or without suicidal tendencies and irritability (five countries). In China, somatic, sleep, appetite and fatigue indicators loaded onto the third factor. Further CFAs with a two-factor structure (affective suffering and motivation) did not yield an acceptable fit to the data.

Using SHARE wave 5 data, Portellano-Ortiz et al. (2018) found similar results to those in Prince et al. (1999) and Castro-Costa et al. (2007). The Exploratory Factor Analysis (EFA) found an affective suffering and a (lack of) motivation factor, but the item "suicidality" was not included in either factor.

Maskileyson et al. (2021) examined the factor structure using SHARE wave 6 data across 17 European countries and Israel. The EFA solution replicated the two aforementioned factors of affective suffering and lack of motivation, but the suicidality, lack of appetite, sleep and fatigue items were not included in the following multi-group analyses because they either failed to load substantially on any factor or loaded on both factors in more than 25% of the countries. The multigroup CFA further showed that the scale was not invariant across countries.

Tomás et al. (2022) analyzed the EURO-D using SHARE wave 8 data. Their CFA supported a bifactor structure with a general depression factor and two specific factors (affective suffering and lack of motivation). They also examined Differential Item Functioning (DIF) by gender and found no substantive item bias between men and women. Finally, two recent studies tested the longitudinal invariance of the EURO-D scale and confirmed a one-factor structure for the scale, without scalar invariance across countries. Gana et al. (2023), with waves 1, 2 and 4 of SHARE across the nine European countries who participated in the first wave (Austria, Belgium, Denmark, France, Germany, Italy, Spain, Sweden and Switzerland) and Fong and Chan (2025) with the 27 European countries and Israel that participated in waves 8 and 9 of the SHARE project.

Regarding the factor structure, Table 1 summarizes the information from these validation studies of the EURO-D scale.

Reflecting on the aforementioned literature, it seems that the structure of the EURO-D scale remains unclear and, therefore, needs further scrutiny. Potential explanations of this lack of concordance are the nature of the techniques employed in the study of dimensionality (exploratory and confirmatory), as well as the type of model employed (first-order confirmatory models and bi-factor models). In particular, the many subjective decisions that are involved in the EFA may obscure the structure of the scale, as previous simulation studies have reported (Garrido & Abad, 2013; Lübbe, 2019).

Although we have seen that there are several exploratory statistical techniques, including Exploratory Factor Analysis or Exploratory Structural Equation Models (ESEM), it is worth exploring alternative approaches to analyze the dimensionality of the EURO-D. One such alternative analytical strategy is the Exploratory Graph Analysis (EGA), a recently developed technique that stems from network psychometrics (Golino & Epskamp., 2017). Network psychometrics applies network modeling techniques to analyze the dimensionality of scales and questionnaires (Epskamp et al., 2015). The EGA employs the Gaussian graphical model (GGM) (Lauritzen, 1996; Epskamp et al., 2017) to estimate the joint distribution of random variables. Nodes or, in the case of scales, items are connected by edges (associations) based on partial correlation coefficients (Golino et al., 2020). This is latter followed by clustering techniques to detect the dimensionality in the network (Laskowski et al., 2023).

There is evidence from simulation studies already showing that EGA is able to capture dimensionality more adequately than the usual exploration methods used in factor analysis and principal component analysis. Golino and Epskamp (2017) simulated 32,000 datasets by manipulating the number of factors, number of items, sample size and correlation between factors. They found that the

**Table 1.** EURO-D scale validation studies

| Authors | Year | Countries | Sample size | Method | Factor structure | Item loads Affective Suffering | Motivation |
|---|---|---|---|---|---|---|---|
| Prince et al. | 1999 | 11 European countries | 21,724 | PCA | Two-factors | I1; I2; I3; I7; I8; I9 | I6; I10; I11 |
| Castro-Costa | 2007 | 10 European countries | 22,777 | CFA | Two-factors | I1; I3; I4; I5; I7; I9; I12 | I2; I6; I10; I11 |
| Guerra et al. | 2015 | 9 countries | 17,852 | PCA | Two or three factors | I1; I3; I4; I5; I7; I9; I12 | I2; I6; I10; I11 |
| Portellano-Ortiz et al. | 2018 | 17 European countries and Israel | 63,755 | EFA | Two-factors | I1; I4; I5; I7; I9; I12 | I2; I6; I8; I10; I11 |
| Maskileyson et al. | 2021 | 17 European countries and Israel | 41,862 | EFA and CFA | Two-factors | I1; I4; I5; I7; I12 | I2; I6; I8; I10 I11 |
| Tomás et al. | 2022 | 26 European countries and Israel | 46,317 | CFA | Two-factors | I1; I3; I4; I5; I7; I9; I12 | I2; I6; I8; I10; I11 |
| Gana et al. | 2023 | 9 European countries | T1 = 21,639, T2 = 21,791, T3 = 32,360, T4 = 42,346, T5 = 37,159 | TSO and SEM | One-factor | | |
| Fong & Chan | 2025 | 27 European countries and Israel | 38,047 | CFA | One-factor | | |

*Note*: CFA = confirmatory factor analysis; EFA = exploratory factor analysis; I = item; I1 = depressive mood; I2 = pessimism; I3 = suicidality; I4 = guilt; I5 = trouble with sleep; I6 = lack of interest; I7 = irritability; I8 = lack of appetite; I9 = fatigue; I10 = concentration problems; I11 = lack of enjoyment; I12 = tearfulness; PCA = principal component analyses; SEM = structural equation model; T = time; TSO = Trait–State-Occasion model.

EGA outperformed other exploratory methods to reduce dimensionality, specifically parallel analysis (PA), Kaiser–Guttman's eigenvalue greater-than-one rule, multiple average partial procedure (MAP), very simple structure (VSS) and maximum-likelihood approaches. In addition, Golino et al. (2020) further manipulated items' factor loadings and skewness and analyzed the suitability of a new algorithm for the EGA. Their results again pointed to the outperformance of EGA against alternative methods.

Given these promising results of EGA for analyzing the dimensionality of scales and questionnaires, together with the need to further explore the dimensionality of the EURO-D, the main aim of this study is to explore the dimensionality of the EURO-D with EGA, and to confirm this dimensionality using CFA.

## Method

### Sample and procedure

In this study, we employed data from the 8th wave of SHARE (Börsch-Supan et al., 2013; SHARE-ERIC, 2024), the most recent wave at the time of the start of this study, collected in 2019 and 2020. SHARE is targeted at individuals aged at least 50 years and their partners, irrespective of their age. Currently, SHARE has nine waves of data that have been collected biannually since the start of the project in 2004. The sampling strategy follows a probabilistic approach with slight variations across participating countries; more details on the sampling strategy are available in Bethmann et al. (2019). The SHARE study meets ethical criteria for research and has been reviewed and approved by the Ethics Council of the Max Planck Society (for an overview, please see: https://share-eric.eu/fileadmin/user_upload/Ethics_Documentation/SHARE_ethics_approvals.pdf).

The sample employed in the study comprised 45,390 individuals from 26 European countries who were 50 years or older. Although the original sample from the eight waves of the SHARE project

includes data from Israel, we excluded this country from our analyses based on geopolitical and ethical considerations. Geopolitically, our study is framed as European in scope, and we therefore selected countries with broadly comparable sociopolitical, welfare, and health-care system contexts. From an ethical perspective, recent scholarly work has highlighted the need for ethical and epistemological caution when working with data connected to contexts of ongoing large-scale violence (Bdier et al., 2025; Ziadah et al., 2025). The mean age was 71.27 years (SD = 9.35), with ages ranging from 50 to 104 years. Additional sociodemographic characteristics are provided in Table 2.

### Instruments

The EURO-D scale (Prince et al., 1999) consists of 12 items tapping the following depressive symptoms: depressed mood, pessimism, suicidal ideation, guilt, sleep, lack of interest, irritability, lack of appetite, fatigue, concentration problems, lack of enjoyment and crying. The scale score ranges from 0 (no depressive symptomatology) to 12 (maximum depressive symptomatology) (Mehrbrodt et al., 2019).

### Statistical analysis

First, we computed descriptive statistics of the items of EURO-D. Additionally, factor structure was analyzed with both EGA and CFA. In order to do so, the overall sample was randomly divided into two samples of 22,823 and 22,567 people, respectively. The random samples were sampled from the overall sample with the random number generator in SPSS 28. In the first sample (derivation sample), EGAs were applied to explore the dimensionality of the EURO-D scale, while in the second sample (cross-validation sample), the structures found in the EGAs were tested using CFA to examine model fit of these structures to the data.

Golino and Epskamp (2017) proposed EGA to estimate dimensions or latent variables in a dataset (usually items of a scale) using

**Table 2.** Sociodemographic characteristics of the participants

| Characteristics | | *N* (%) or mean (SD) | |
|---|---|---|---|
| Women | | 26,040 (57.4%) | |
| Age | | 71.27 (9.35) | |
| Years of education | | 11.22 (4.15) | |
| Married, living with partner | | 26,627 (67.5%) | |
| Married, not living with partner | | 421 (1.1%) | |
| Never married | | 2016 (5.1%) | |
| Divorced | | 3,281 (8.3%) | |
| Widowed | | 7,067 (17.9%) | |
| Country | % | Country | % |
| Austria | 1,561 (3.4%) | Croatia | 1,182 (2.6%) |
| Belgium | 1995 (4.4%) | Greece | 2,981 (6.6%) |
| France | 2,475 (5.5%) | Italy | 2,151 (4.7%) |
| Germany | 2,872 (6.3%) | Malta | 789 (1.7%) |
| Luxemburg | 948 (2.1%) | Slovenia | 2,484 (5.5%) |
| Netherlands | 1921 (4.2%) | Spain | 2094 (4.6%) |
| Switzerland | 1895 (4.2%) | Cyprus | 525 (1.2%) |
| Denmark | 2,164 (4.8%) | Bulgaria | 891 (2.0%) |
| Estonia | 3,013 (6.6%) | Czech Republic | 2,701 (6.0%) |
| Finland | 1,153 (2.5%) | Hungary | 775 (1.7%) |
| Latvia | 771 (1.7%) | Poland | 2051 (4.5%) |
| Lithuania | 1,415 (3.1%) | Romania | 1,249 (2.8%) |
| Sweden | 2,354 (5.2%) | Slovakia | 980 (2.2%) |

undirected network models (1996). The EGA estimates a network using the GGM (1996) and then applies a clustering algorithm for weighted networks (2020). In the R-package EGAnet, two algorithms are available to estimate the level of sparsity in the GGM: the Graphical Least Absolute Shrinkage and Selection Operator (GLASSO; Massara et al., 2016), and the Triangulated Maximally Filtered Graph (TMFG; Friedman et al., 2008). Both algorithms will be applied to the derivation sample.

The aim of CFA is to test whether a hypothesized measurement model, which may be based on theory and/or previous analytic research, fits a given dataset. In our context, CFAs are estimated to test the structures found in the two EGAs and to analyze their relative model fit. Model fit of the CFAs was examined with the standard goodness-of-fit indexes and statistics: the chi-square statistic ($\chi^2$), the comparative fit index (CFI), the root mean square error of approximation (RMSEA) and the squared root mean residual (SRMR) and the NNFI (non-normed fit index). According to Hu and Bentler (1999), CFI values ≥.90 and RMSEA and SRMR values ≤.08 indicate acceptable fit, while values of at least .95 for CFI and NNFI, no more than .05 for RMSEA and SRMR, are considered as indicating excellent fit. CFAs were estimated using diagonally weighted least squares (DWLS), given the binary nature of the data (Finney and DiStefano, 2006).

All analyses were performed in R (R Core Team, 2021), the EGAs in the package EGAnet and the CFAs in the package lavaan (Rosseel, 2012).

## Results

### *Descriptive statistics*

Table 3 shows percentages of positive responses to each of the 12 items from EURO-D in the overall sample. Indicators of depressive symptomatology with a higher occurrence in the sample were sleeping problems, fatigue and tearfulness.

### *Exploratory graph analysis: derivation sample*

Figure 1 presents the graphical network with two latent dimensions when GLASSO estimation is used. This graph suggests two dimensions for the EURO-D, with the thickness of the edges representing the strength of correlations. The color of the edges further indicates positive associations among items, given that these are all green. Closeness of the items (nodes) represents the degree of relationship to each other. First dimension in the graph mostly groups items referred to in other validation studies as affective suffering, while the second dimension groups items of lack of motivation or simply motivation. Loadings for all items are presented in Table 4.

We used the bootEGa function to assess the stability of the dimensionality in the GLASSO solution. Bootstrap replications suggested two dimensions in 85.0% of the occasions and three in the rest, indicating good replicability. Additionally, Figure 2 shows how many times (in percentage) items were replicated in the same dimension. In the GLASSO estimation, the items in the dimension of affective suffering were replicated very well, but some items (nodes) in the dimension of lack of motivation had some problems of replication, specifically lack of appetite, suicidality and fatigue. No surprise that these items are at the edge of both dimensions in Figure 1.

Figure 3 presents the graphical network with two latent dimensions when TMFG estimation is used. Again, the network suggests two dimensions for EURO-D. First dimension in the graph mostly groups items referred to in other validation studies as affective suffering, while the second dimension groups items of lack of motivation. Loadings for all items are presented in Table 2.

We then again used the bootEGa function to assess the stability of the dimensionality in the GLASSO solution. Bootstrap replications suggested 84.8% suggested two dimensions, with 14.4% suggesting three, and a residual 0.8% suggesting four, which is a very good replicability of the two dimensions. Additionally, Figure 4

**Table 3.** Percentages of response for the items

| Item (yes) | Yes (%) |
|---|---|
| Depressive mood | 40 |
| Pessimism | 18 |
| Suicidality | 6 |
| Guilt | 7.5 |
| Trouble with sleep | 36.7 |
| Lack of interest | 10.8 |
| Irritability | 25.8 |
| Lack of appetite | 9.4 |
| Fatigue | 35.6 |
| Concentration problems | 17.2 |
| Lack of enjoyment | 13.8 |
| Tearfulness | 23.7 |

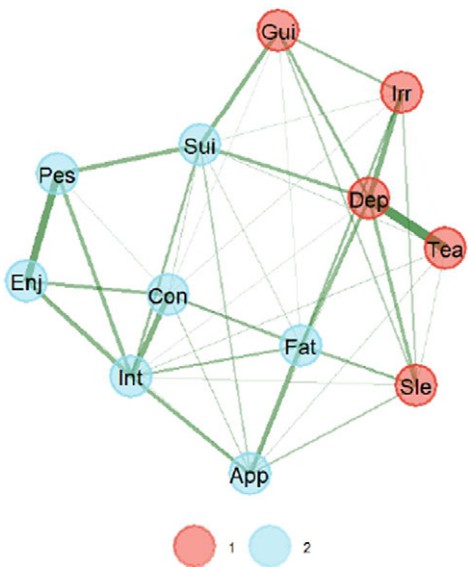

**Figure 1.** Graphical network with the GLASSO estimation method. *Note*: The thickness of the edge is the degree of correlation, with positive correlations depicted as green. A strong correlation is represented by the closeness of the items. *Note*: 1 = affective suffering, 2 = lack of motivation.

**Table 4.** Loadings for the EGAs and the CFAs

| | EGA GLASSO | | EGA TMFG | | CFA-GLASSO | | CFA-TMFG | |
|---|---|---|---|---|---|---|---|---|
| Items | F1 | F2 | F1 | F2 | F1 | F2 | F1 | F2 |
| Depression | .704 | | 1.447 | | .837 | | .793 | |
| Tearfulness | .347 | | .736 | | .666 | | .643 | |
| Irritability | .320 | | .931 | | .566 | | .548 | |
| Guilt | .263 | | .800 | | .507 | | .491 | |
| Sleep | .254 | | .673 | | .593 | | .575 | |
| Interest | | .607 | | 1.094 | | .731 | | .790 |
| Concentration | | .418 | | .992 | | .624 | | .670 |
| Pessimism | | .430 | | .872 | | .485 | | .526 |
| Enjoyment | | .421 | | 1.007 | | .523 | | .567 |
| Suicidality | | .315 | 1.112 | | | .707 | .696 | |
| Appetite | | .336 | | .805 | | .642 | | .680 |
| Fatigue | | .312 | .979 | | | .700 | .672 | |

*Note*: Only the largest loadings on each factor for the EGAs are shown for clarity; CFA = confirmatory factor analysis; EGA = exploratory graph analysis; F1 = Factor 1; F2 = Factor 2; GLASSO = Graphical Least Absolute Shrinkage and Selection Operator; TMFG = Triangulated Maximally Filtered Graph.

shows how many times (in percentage) items were replicated in the same dimension as the TMFG solution. Appetite was the only item with some replicability problems. No surprise that this item is on the edge of both dimensions in Figure 3.

It is worth noting that the items that are changing dimensions from one solution to the other are fatigue and suicidality. To determine which EGA solution is better, we calculated the Total Entropy Fit Index using Von Neumman's entropy for correlation matrices (TEFIvn). Lower values suggest a better fit of a structure to the data (Golino et al., 2021), and the TEFIvn for GLASSO solution was −6.61, while TEFIvn for TMFG was −6.65, slightly lower but very similar.

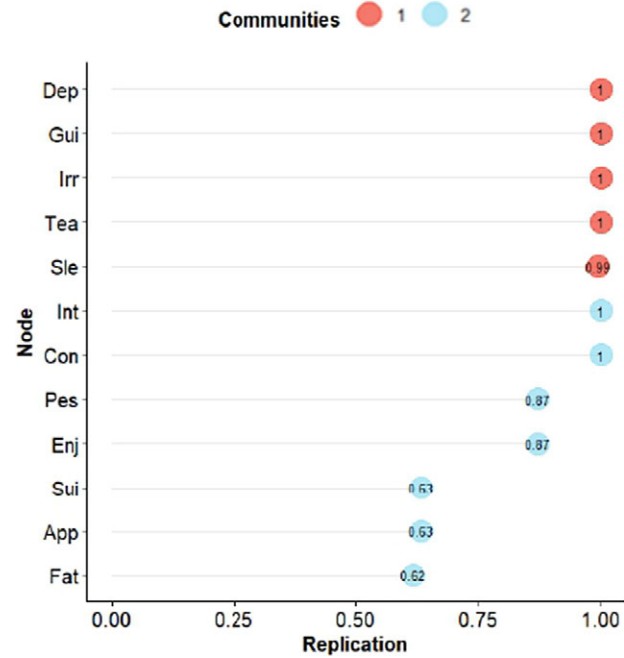

**Figure 2.** Percentages of bootstrapped replicates of each indicator in its cluster (dimension) for GLASSO estimation. *Note*: 1 = affective suffering, 2 = lack of motivation.

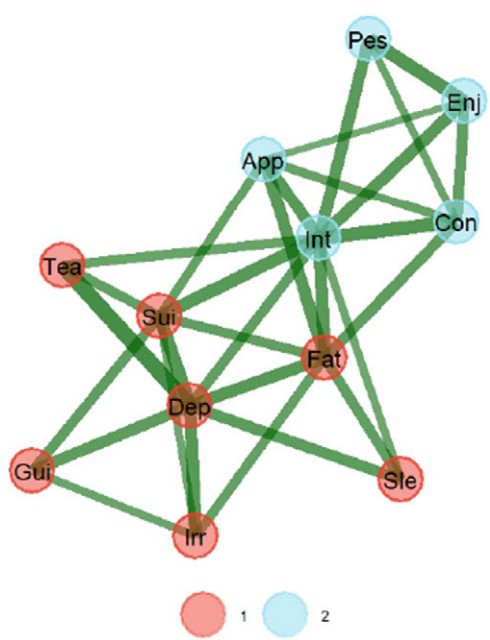

**Figure 3.** Graphical network with the TMFG estimation method. *Note*: The thickness of the edge is the degree of correlation, with positive correlations depicted as green. A strong correlation is represented by the closeness of the items. *Note*: 1 = affective suffering, 2 = lack of motivation.

### Confirmatory factor analyses: Cross-validation sample

Once the exploratory dimensionality of the scale had been analyzed with the two estimation procedures in EGA, the validation sample was employed to confirm these two structures. Therefore, two CFAs were specified and tested, corresponding to the structures in the GLASSO and TMFG EGAs. Both CFAs fitted the data well. On one hand, the CFA based on the dimensionality found in the GLASSO

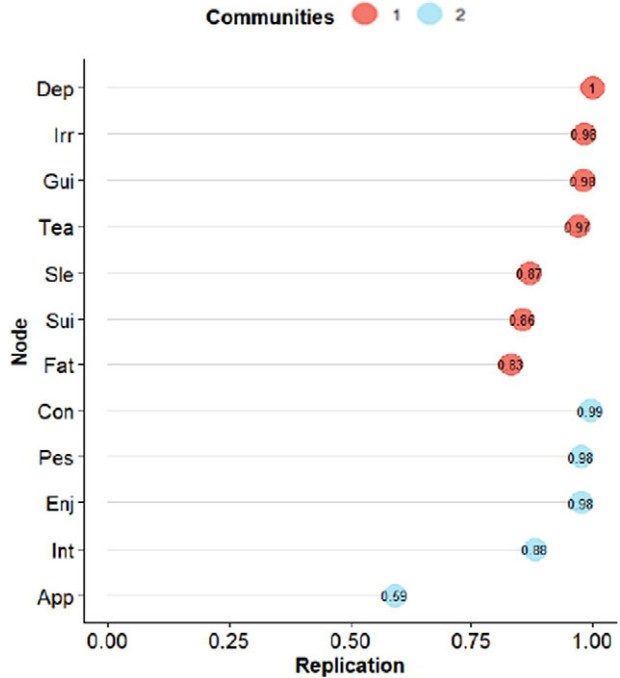

**Figure 4.** Percentages of bootstrapped replicates of each indicator in its cluster (dimension) for TMFG estimation. *Note*: 1 = affective suffering, 2 = lack of motivation.

EGA showed the following results: $\chi^2$ (53) = 2,472.71, $p$ < .001, CFI = .941, NNFI = .958, RMSEA = .045, 90% CI [.044, .047], SRMR = .061. Standardized factor loadings are displayed in Table 2. On the other hand, the CFA based on the dimensionality of the TMFG EGA showed the following results: $\chi^2$ (53) = 2,356.17, $p$ < .001, CFI = .944, NNFI = .960, RMSEA = .044, 90% CI [.043, .046], SRMR = .061. Again, standardized factor loadings can be consulted in Table 2, and the correlation between the latent factors was .695 ($p$ < .001).

Overall, the results of the CFA are in line with those of the EGAs; that is, the two dimensions found exploratorily have been confirmed, with a slightly better fit in the CFA based on the results of the TMFG EGA.

## Discussion

Results from the EGA have shown the EURO-D displays a two-factor structure of affective suffering and lack of motivation using both the GLASSO and the TMFG estimation methods. Previous studies (Guerra et al., 2015; Maskileyson et al., 2021; Tomás et al., 2022) identified a similar factor structure with consistent item-factor relationships. In line with these findings, our analysis using both estimation methods showed that items related to depression, tearfulness, irritability, guilt and sleep loaded onto the affective suffering dimension, while items concerning interest, pessimism, enjoyment and concentration loaded onto the lack of motivation dimension.

However, small deviations in factor structure were observed between the estimation methods. More concretely, the suicidality and fatigue items loaded onto the lack of motivation factor in the GLASSO method and onto the affective suffering dimension in the TMFG method. Previous studies examining the factor structure of this scale have also reported problems with these two items, as well as with the lack of appetite item (Castro-Costa et al., 2007; Guerra

et al., 2015; Portellano-Ortiz et al., 2018; Maskileyson et al., 2021; Tomás et al., 2022).

In general terms, previous studies report the suicidality and fatigue items to load onto the affective suffering factor (Prince et al., 1999; Castro-Costa et al., 2007; Guerra et al., 2015; Tomás et al., 2022). Results from Portellano-Ortiz et al. (2018) allocate the fatigue item onto the affective suffering dimension but drop the suicidality item. On their part, Maskileyson et al. (2021) reported fatigue and suicidality items to cross-load and thus recommended dropping both items from the scale. Finally, although loading onto the affective suffering dimension, Tomás et al. (2022) warned about the small loading size of these items. All in all, evidence seems to provide stronger support for the EGA solution found with the TMFG estimation method.

Regarding lack of appetite, in the present study, this item was consistently allocated to the lack of motivation, but in the TMFG solution, appetite presented problems of replicability. Prince et al. (1999) reported the appetite item to load onto affective suffering, while some studies decided to drop the item given severe cross-loadings (Castro-Costa et al., 2007; Guerra et al., 2015; Maskileyson et al., 2021). In turn, results from Portellano-Ortiz et al. (2018) and Tomás et al. (2022) support the allocation of lack of appetite into lack of motivation, but the latter also warned about the small size of this item's loading onto the factor.

Turning to confirmatory models, model fit results indicated that both structures fit the data adequately. Be that as it may, the model based on TMFG estimation displayed a slightly better fit. That, together with previous literature allocating suicidality and fatigue in the affective suffering dimension (Prince et al., 1999; Castro-Costa et al., 2007; Guerra et al., 2015; Tomás et al., 2022), seems to favor this latter model.

Most of the previous studies (Prince et al., 1999; Castro-Costa et al., 2007; Portellano-Ortiz et al., 2018) on the factor structure of the EURO-D scale used PCA as an exploratory technique to identify latent factors of depression. However, it is widely known that PCA is not a method of factor analysis (FA) but one aimed at reducing dimensionality (Widaman, 1993). As argued by Widaman (2018), PCA was developed to explain manifest variables' variance by combining them into components with certain properties, while EFA is aimed at explaining the correlations among manifest variables under the assumption that an underlying latent variable or factor can account for such correlations.

In this study, we employed EGA, a recently developed technique that estimates relationships among observed or manifest variables. When estimating the relationship between a certain pair of variables, it is done conditioned on the rest of the variables. When several variables are related to each other, they form a cluster that accounts for a latent variable (Golino and Epskamp, 2017). In contrast to EFA, EGA is an automated process that does not require researchers to make decisions such as how many factors to retain or which rotation method to employ.

In addition, this study employed a cross-validation strategy, which allowed proving the structure found by EGA using a confirmatory technique within the Structural Equation Modeling (SEM) framework. Previous work using this strategy (Portellano-Ortiz et al., 2018) used PCA to establish the factor structure of the EURO-D and later employed CFA to validate it. According to Widaman (2018), PCA does not provide a robust basis for SEM.

However, this study also has several limitations. Since the development of the scale, different factor structures have been identified across samples from different countries, defying cross-cultural invariance (Prince et al., 1999; Guerra et al., 2015; Maskileyson et al., 2021). In this line, studies reporting the same structure as the

one found in this work employed data from previous waves of SHARE (Portellano-Ortiz et al., 2018; Tomás et al., 2022). Therefore, the diversity of factor solutions reported in the literature may respond to cross-country differences. Across countries, depression scales load onto different factors and show differential item functioning, indicating that structural inequalities and health-system barriers shape how depressive symptoms are experienced, expressed and measured (Castro-Costa et al., 2008; Bergenfeld et al., 2023; Yu et al., 2024). Structural contexts also shape the identification of depressive symptomatology and the delivery of care (Carbonell et al., 2020).

Finally, when comparing depression scales across countries, we should also consider language and specific wording. In this regard, our study used data from 26 European countries, most of which have different languages. Previous studies have shown that the EURO-D scale is influenced by language and culture. For example, Guerra et al. (2015) found measurement invariance in Latin American and Indian samples, but not in Chinese or Nigerian samples. However, a recent study testing the measurement invariance of the EURO-D scale using SHARE data found that an alignment-based measurement invariance model demonstrated acceptable model fit indices across European countries (Fong and Chan, 2025).

Future studies should continue updating the measurement invariance of the EURO-D across countries because, with increasing globalization and economic and human development, more cultures may gradually converge toward a consensus of depression symptomatology.

## Conclusions

This study contributes to the growing body of evidence on the factor structure of the EURO-D by using a combination of an exploratory technique, EGA, and a confirmatory technique, CFA. The results support a two-factor structure on the EURO-D scale composed of "affective suffering" and "lack of motivation" as in previous studies. The results also showed different assignments of the fatigue and suicidality items to the two EURO-D dimensions. These results underscore the utility of the EGA as a tool for investigating dimensionality, particularly in scales such as EURO-D, where factor structures remain unclear.

**Open peer review.** To view the open peer review materials for this article, please visit http://doi.org/10.1017/gmh.2026.10204.

**Data availability statement.** The data used for this article can be accessed at the SHARE Research Data Center after registration (www.share-project.org).

**Acknowledgments.** The SHARE data collection has been funded by the European Commission DG RTD through FP5 (QLK6-CT-2001-00360), FP6 (SHARE-I3: RII-CT-2006-062193, COMPARE: CIT5-CT-2005-028857, SHARE-LIFE: CIT4-CT-2006-028812), FP7 (SHARE-PREP: GA N°211909, SHARE-LEAP: GA N°227822, SHARE M4: GA N°261982, DASISH: GA N° 283646) and Horizon 2020 (SHARE-DEV3: GA N°676536, SHARE-COHESION: GA N°870628, SERISS: GA N°654221, SSHOC: GA N°823782, SHARE-COVID19: GA N°101015924), and by DG Employment, Social Affairs and Inclusion through VS 2015/0195, VS 2016/0135, VS 2018/0285, VS 2019/0332, VS 2020/0313 and SHARE-EUCOV: GA N°101052589 and EUCOVII: GA N°101102412. Additional funding from the German Ministry of Education and Research, the Max Planck Society for the Advancement of Science, the U.S. National Institute on Aging (U01_AG09740-13S2, P01_AG005842, P01_AG08291, P30_AG12815, R21_AG025169, Y1-AG-4553-01, IAG_BSR06-11, OGHA_04-064, BSR12-04, R01_AG052527-02, HHSN271201300071C, RAG052527A) and from various national funding sources is gratefully acknowledged (see www.share-eric.eu).

**Author contribution.** Conceptualization: JMT. Methodology: MA and IF. Formal analysis, MA, JMT and IF. Writing – original draft preparation: MA and IF. Writing – review and editing: ZT and AS. Visualization: ZT. Supervision: JMT and IF. Funding acquisition: JMT. All authors reviewed the results and approved the final version of the manuscript.

**Financial support.** This work was supported by MCIN/AEI/10.13039/501100011033 and by "ERDF A way of making Europe," framed in the project PID2021-124418OB-I00.

**Competing interests.** The authors declare none.

**Ethics statement.** Data for this study were reviewed and approved by the Ethics Council of the Max Planck Society. For an overview and summary of the ethics approvals, please see: https://share-eric.eu/fileadmin/user_upload/Ethics_Documentation/SHARE_ethics_approvals.pdf. All participants freely consented to their participation in the Survey of Health, Ageing and Retirement in Europe.

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
