## [Reviewer Report]

Thank you for the opportunity to review the manuscript titled “Revisiting the factor structure of the EURO-D, a depressive symptomatology scale: an Exploratory Graph Analysis in 26 European countries and Israel.” The study aims to validate the factor structure of the EURO-D scale using Exploratory Graph Analysis (EGA) and confirmatory factor analysis. I identify several areas that would benefit from further improvement. My comments are detailed below.

1) Table 1 could be enhanced by including an additional column indicating which items load onto each dimension in each study reviewed. This would improve clarity and facilitate comparison among the presented findings.

2) Geopolitical and ethical considerations in sample selection must be addressed. In the introduction, the authors state that their estimates and comparisons are based on “European” data. However, the dataset includes Israel, a country that is not part of Europe. The authors should clarify the criteria used to define the analytical region. At present, the inclusion of Israel is methodologically inconsistent and introduces systematic bias in cross-country comparability. While it is understood that SHARE includes Israel among its participating countries, this alone does not justify its inclusion in a study that is framed as European in scope. This is not merely a geographic imprecision, but a substantive issue affecting the interpretability of results.

As demonstrated in extensive literature, mental health indicators are strongly shaped by political, structural, and conflict-related determinants. Therefore, selecting countries that do not share comparable sociopolitical, welfare, and healthcare system configurations may distort factor structures and prevalence estimates.

Additionally, and of particular relevance at this time, there is a growing body of scholarly work addressing the current situation in Gaza as one of systematic, large-scale violence against the Palestinian population, with direct implications for health, psychological well-being, and social conditions (Bdier et al., 2025; Children’s Geographies Editorial Board, 2024; Ziadah et al., 2025, etc.). This literature calls for ethical and epistemological responsibility on the part of the international academic community when dealing with data connected to ongoing mass violence.

Therefore, I strongly recommend:

• Removing Israel from the dataset and re-running the analyses to ensure conceptual and methodological coherence with the stated regional scope (Europe).

• Explicitly justifying the regional boundaries of the study in the Methods section.

• Including a brief reflection in the Limitations section acknowledging the ethical responsibility of analytical choices and explaining the exclusion of this country.

3) In the Sample and Procedure section, the authors state that they used data from wave 8 of SHARE because it was the most recent available at the time of the study. However, it would be helpful to clarify which waves are included in SHARE overall, to provide a clearer understanding of the data framework.

4) I would appreciate further clarification on the decision to use Exploratory Graph Analysis (EGA) rather than other more integrative and flexible approaches, such as Exploratory Structural Equation Modeling (ESEM). The rationale provided refers to inconsistency in previous findings across exploratory and confirmatory methods. Given this concern, ESEM may have been particularly appropriate, as it explicitly incorporates elements of both.

5) Minor errors should be corrected. In the first paragraph of the Introduction, there are inconsistencies such as “de-pressed mood” and “de-pressive.” In the second paragraph, the citations for Massara et al. (2016) and Friedman et al. (2008) appear in font size 11, while the rest of the manuscript is in size 12.

6) In the Limitations section, when discussing the lack of cross-cultural invariance, it would be valuable to reference literature linking such differences to structural inequalities and barriers in mental healthcare systems. For instance, Carbonell (2020) examines how structural contexts shape both the identification of depressive symptomatology and the delivery of care. Integrating this perspective would expand the interpretive framework beyond psychometric considerations.

7) Finally, I suggest revising some passages where the writing is repetitive, particularly in the historical–methodological background section, in order to improve clarity, avoid conceptual redundancy, and enhance the overall readability of the manuscript.

References

Abuward, O. H., Fiscone, C., Abulibda, N. M., Mustafa, A., & Veronese, G. (2025). Do you think all of us are mentally sick because of this horror? Mental health under genocide in Gaza. International Journal of Human Rights in Healthcare, 18(4), 225-243.

Bdier, D., Hamamra, B., & Mahamid, F. (2025). From laughter to survival: The effect of war on children’s play in Gaza. Children and Youth Services Review, 108526.

Carbonell, A., Navarro‐Pérez, J. J., & Mestre, M. V. (2020). Challenges and barriers in mental healthcare systems and their impact on the family: A systematic integrative review. Health & Social Care in the Community, 28(5), 1366-1379.

Children’s Geographies Editorial Board. (2024). Editorial statement on violence against children and a call for a ceasefire in Gaza. Children’s Geographies.

Ziadah, R., Henderson, C., Jabary Salamanca, O., Plonski, S., Chua, C., Al Sanah, R., & El Khazen, E. (2025). Disruptive Geographies and the War on Gaza: Infrastructure and Global Solidarity. Geopolitics, 1-39.

---

## [Reviewer Report]

Overall, the manuscript is appropriately prepared, timely, and valuable. It contributes new knowledge regarding the dimensionality of a widely used instrument - the EURO-D. I am pleased to have had the opportunity to review this paper.

Nevertheless, although the manuscript is generally well written, I have three comments:

1. My primary concern is that the analyses are conducted on a pooled cross-national sample. The sources cited in the Introduction (pp. 4-5) clearly indicate that country-level differences are important. When working with SHARE data, this issue is further complicated by the fact that the data involve not only multiple countries but also multiple languages - some countries administer more than one language version. In the current version of the manuscript, this issue is only briefly mentioned at the end of the Discussion (pp. 15–16). At a minimum, I would expect the authors to expand the limitations section and include a discussion of how the applied statistical methods may be affected in a multilingual sample. If feasible, I would also recommend considering measurement invariance testing or another procedure that explicitly addresses this concern.

2. I would also encourage the authors to account for the fact that the EURO-D items are dichotomous and that respondents may rarely endorse certain symptoms. Table 3 indicates that suicidality was endorsed by 6% of the sample, and guilt by 7.5%. The authors should clarify whether such low endorsement rates could have influenced the results, particularly given evidence that, at least in simulated non-dichotomous data, skewness can affect EGA performance (Markos & Tsigilis, 2024, https://doi.org/10.3389/fpsyg.2024.1359111).

3. A minor point: it would be helpful for Table 2 to report the absolute frequencies (N) in addition to percentages, as this would make it easier for readers to interpret the context.

---

## [Editor Report]

Thank you for submitting this research. As noted from the reviewers, there is strong value in the work you are presenting but several areas that need further clarification and potentially additional analyses. In particular, given the global nature of this journal, it will be very important to explore the within-European variation in the scale - recognizing that the populations across the different countries may be sufficiently distinct that simply pooling the data is insufficient to fully explore the factor structure.

---

## [Reviewer Report]

I acknowledge the considerable effort undertaken by the authors in revising the manuscript, particularly in reorienting the analyses in line with more stringent ethical standards. This decision is especially noteworthy, as it has entailed a substantive reformulation of the original work, thereby strengthening the scientific integrity of the study.

In its current form, the manuscript presents a more coherent structure and an adequately developed methodological and analytical framework, in line with expected scholarly standards. However, an inconsistency remains in the abstract, which does not accurately reflect the changes introduced in the methodological approach and the resulting findings. A revision of the abstract is therefore required to ensure internal coherence.

Subject to this minor revision, the manuscript is recommended for acceptance.

---

## [Reviewer Report]

Thank you to the authors for the targeted revisions. All of my comments have been addressed, and I have no further remarks..

---

## [Editor Report]

authors have revised the remaining issue - the abstract - and the manuscript is now appropriate for acceptance.